# Fibroid Removal after Myomectomy: An Overview on the Problems of Power Morcellation

**DOI:** 10.3390/healthcare10102087

**Published:** 2022-10-19

**Authors:** Radmila Sparić, Mladen Andjić, Ottavia D’Oria, Ivana Babović, Zagorka Milovanović, Gaetano Panese, Martina Licchelli, Đina Tomašević, Andrea Morciano, Andrea Tinelli

**Affiliations:** 1Clinic for Gynecology and Obstetrics, University Clinical Centre of Serbia, Faculty of Medicine, University of Belgrade, 11000 Belgrade, Serbia; 2Clinic for Gynecology and Obstetrics, University Clinical Centre of Serbia, 11000 Belgrade, Serbia; 3Department of Medical and Surgical Sciences and Translational Medicine, PhD Course in “Translational Medicine and Oncology”, Sapienza University, 00185 Rome, Italy; 4Clinic for Gynecology and Obstetrics, Narodni Front, Faculty of Medicine, University of Belgrade, 11000 Belgrade, Serbia; 5Department of Obstetrics and Gynecology and CERICSAL (CEntro di RIcerca Clinico SALentino), “Veris delli Ponti Hospital”, Scorrano, 73020 Lecce, Italy; 6Department of General Surgery and Pediatric Surgery, General Hospital “Čačak”, 32000 Čačak, Serbia; 7Department of Obstetrics and Gynecology, “Cardinal Panico” Hospital, 73039 Tricase, Italy

**Keywords:** uterine fibroid, morcellation, occult leiomyosarcoma, open myomectomy, laparoscopic myomectomy

## Abstract

The authors reviewed uterine fibroid (UF) morcellation and its potential consequences, notably a hypothetical spread and dissemination of occult uterine leiomyosarcoma (LMS) tissue, evaluating the effect of laparoscopic versus open myomectomies with and without morcellation on patients’ outcomes, as well as related medical-legal issues. MEDLINE and PubMed search was performed for the years 1990–2021, using a combination of keywords on this topic. Relevant articles were identified and included in this narrative review. There is an individual risk, for all patients, for LMS diagnosis after myomectomy. However, the risk for occult LMS diagnosis during a laparoscopic myomectomy is generally reduced when the guidelines of scientific societies are followed, with an overall benefit from the laparoscopic approach with morcellation in appropriate cases. Gynecological societies do not ban morcellation and laparoscopic hysterectomy/myomectomy per se, but recommend their use on the basis of the patients’ clinical characteristics. It is suggested for gynecologists to provide detailed information to patients when obtaining an informed consent for open or laparoscopic hysterectomy/myomectomy. A detailed preoperative assessment of patients and the risk benefit ratio of laparoscopic morcellation of uterine mass could overcome the “a priori” banning of the morcellation technique.

## 1. Introduction

Power morcellation was approved by the Food and Drug Administration (FDA) in 1995, as a primary method used for removal of uterine fibroids (UF) during laparoscopic myomectomy [1]. By this procedure, electrical energy is transformed into mechanical power, cutting uteri or fibroids into smaller pieces, for removal from the abdominal cavity via a 12–20 mm ancillary port [2]. The power morcellation techniques underwent technological advances, but none of these have a significant influence on device utility [3]. Performance of morcellation was “uncontained” till to 2014, with the removal of uterine or fibroid fragments from the pelvis and abdomen without problems. Then, after an FDA ban, it was translated, in United States of America (USA) from 2014, into a “contained” morcellation in a retrieval bag, inserted in one of the trocar sites, so that uteri and fibroids are placed within the bag and “in bag” morcellated. Finally, all bags are removed by a mini laparotomic cut at the umbilicus, at the suprapubic area, or at the vaginal posterior fornix, as transvaginal bag extraction [4,5]. This technique showed good results and low incidence of perioperative complications [6].

The FDA decision to ban the “uncontained” morcellation in gynecologic laparoscopic surgery was after the first reports of cancer spread due to an uncontained morcellation [7]. This FDA warning has created bewilderment and perplexity in the world gynecological community, given the overall rarity of the uterine leiomyosarcomas (LMS). It is not only a rare cancer, but hard to diagnose in preoperative assessment; it is prognostically lethal, despite surgical removal. The first FDA statement divided patients who should not undergo morcellation in two groups: the patients who are in “peri or post-menopausal” states and/or who are candidates for “en bloc tissue removal” [8]. Later, the FDA required the use of a morcellator with a containment system, or advised special controls [9].

This narrative review focuses on uterine masses morcellation and its influence on occult LMS tissue spread and dissemination, discussing laparoscopic versus open myomectomies with and without morcellation and evaluating their influence on patients’ outcomes, as well as medico legal issues.

## 2. Materials and Methods

The authors searched the available data on the molecular basis of the pathogenesis, diagnosis, and prognosis of UF and LMS. The authors conducted a MEDLINE, Scopus and PubMed search, for the years 1990–2022, using a combination of keywords, such as “uterine fibroid, “myoma,” “fibromyoma”, “leiomyoma”, “myomectomy,”, “morcellation”, “open myomectomy”, “laparoscopy”, “leiomyosarcoma”, “prognosis”, “treatment”. Literature that was the most relevant to the topic was used based on the authors’ evaluation. Peer-reviewed articles concerning UF, myomas, leiomyomas, LMS and morcellation were included. Additional articles were identified from the references of relevant papers. The terms “uterine fibroids”, “myomas”, “fibromyomas”, and “leiomyomas” can be found in the literature as ways of describing UFs. In the manuscript, we have used the term “fibroid” or “myoma” in equal measure. The results of the research have been divided into sub-sections, in which we illustrate what has been reported in the scientific literature, including the authors’ opinions on such topics.

## 3. Results and Discussion

### 3.1. The Incidence of Unsuspected LMS Tissue in Specimens Retrieved by Myomectomy

Some studies reported results about the incidence of occult unexpected uterine malignancies, especially LMS, in specimens retrieved by morcellation during laparoscopic hysterectomy or myomectomy.

Wright et al. [10] performed an investigation using the prospective database with 232,882 women submitted to laparoscopic hysterectomy, to find the incidence of uterine pathology in women who underwent laparoscopic surgery with morcellators. The primary outcome of the study was to discover the rate of any type of uterine corpus cancers, and the second outcome was to detect malignancies of other parts of the uterus and of adnexal structures, and endometrial hyperplasia. In this study group, 36,470 (15.7%) of women underwent morcellation and among those who underwent morcellation, the prevalence of uterine cancer was 27 per 10,000. The prevalence of other gynecologic malignancies was 7 per 10,000, uterine neoplasms of uncertain malignant potential was 11 per 10,000, and endometrial hyperplasia was 101 cases per 10,000, respectively. According to the results of this study, the authors concluded that advanced age was associated with cancer and endometrial hyperplasia.

Xu et al. [11] conducted research on the incidence of occult uterine malignancies diagnosed after hysterectomy or myomectomy for benign indication. They used cancer registry data and the study included 843 women with occult endometrial carcinoma and 334 women with occult uterine sarcoma. They accessed the registered list of “disease specific” and “generic” cause of mortality, comparing women who underwent laparoscopic supracervical hysterectomy or laparoscopic myomectomy, a surrogate indicator for uncontained power morcellation, with women who underwent supracervical abdominal hysterectomy and total abdominal hysterectomy (TAH), avoiding power morcellation. According to the authors’ results, the uncontained power morcellation was associated with a higher mortality risk in women with occult uterine sarcoma. The incidence and risk for occult uterine LMS is shown in Table 1.

Lieng et al. [20] evaluated the risk of morcellating LMS during laparoscopic supracervical hysterectomy and laparoscopic myomectomy. Basing on immunohistochemical evaluation (Figure 1), they reported that the incidence of uterine LMS in the women diagnosed first with benign fibroids was 0.0054 (1 in 183 women) and the rate of unintended morcellation of a LMS was 0.0002 (1 in 4791 women) during a 13-year period.

Kho et al. [21] analyzed the incidence of occult uterine LMS in 10,119 hysterectomies performed for benign gynecologic indications during the period 2000–2014. The authors showed that occult uterine LMS occurs in 0.089% or just in one case from 1124 hysterectomies for benign gynecologic indications. It is also clear that LMS are not associated with pre-existing uterine leiomyoma [22,23]. Taking into account the low rate of morcellation of occult uterine LMS, the utilization of morcellation must be considered through a benefit to risk ratio for the patient.

Picerno et al. [24] evaluated the incidence of unsuspected uterine sarcoma and other uterine malignancies, and potential malignancies at the time of hysterectomy or myomectomy by power morcellation (Figure 2). The authors performed a retrospective cohort study including 1004 women submitted to laparoscopic hysterectomy or myomectomy by a power morcellator. The primary outcome of the study was the incidence of uterine malignancy and the secondary outcome was the occurrence of other conditions associated with malignant tumors.

They reported two women with uterine malignancy, both endometrial carcinoma (1/502; 95% confidence interval [CI], 1/4144-1/139) and non-uterine sarcomas (97.5% CI, 0-1/273). It was also observed that six (1/167; 95% CI, 1/455-1/77) women showed uterine premalignancy: 2 atypical myomas, 1 STUMP (smooth muscle tumors of uncertain malignant potential), and 3 endometrial atypical hyperplasia. It was observed that women with uterine malignancies and uterine premalignancy had greater uterine weight and, frequently, UF, as indications compared with women with non-uterine malignancies.

Yang et al. [25] reported on a national multicenter study to investigate the proportion of uterine malignant tumors in patients submitted to laparoscopic myomectomy by morcellation. The study was retrospectively performed and included a total of 33,723 patients. The authors postoperatively showed that 62 patients had malignant tumors. The clinical characteristics of these patients showed about 62.9% of these patients with abnormal ultrasonic blood flow signals and 37.1% of these patients with rapid growth of uterus preoperatively.

The results of the studies including a large number of patients showed the low incidence of undiagnosed cancer tissue in UFs tissue specimen retrieval by morcellation. However, the patients with postoperative diagnosed LMS had clinical characteristics which could be evaluated preoperatively and, accordingly, the morcellation could be dismissed in such patients. Unfortunately, there is not available reliable preoperative method to differentiate between a UF and a LMS [26]. The available studies on preoperative differential diagnosis are not sufficient to give us conclusive results, feasible in current clinical practice [26]. It seems that, in the near future, multi-parametric diagnostic scales, including patients’ characteristics, imaging technologies and laboratory tests, could give us promising results in preoperatively differentiating these lesions [26].

Even though the use of morcellation could theoretically lead to the dissemination of malignant cells into the peritoneal cavity, the clear mechanism and degree of dissemination are unknown. Therefore, an open question could be: “Can the UF morcellation potentially cause the undiagnosed LMS cells dissemination during morcellation?”.

Asgari et al. [27] evaluated peritoneal dissemination of spindle cells in laparoscopic and open myomectomies. Their prospective, nonrandomized clinical trial included 150 women suspected of UFs submitted to laparoscopic or open myomectomy. The detection of spindle cells dissemination was performed by washing the peritoneal cavity. The washing was performed after the closure of the myometrial incision and before morcellation and, successively, after morcellation. In the open myomectomy, a washing procedure was performed after the closure of the uterine incision. The dissemination of the spindle cells was found in both the laparoscopic and open myomectomy groups. Therefore, the procedure of morcellation, basing on authors’ research, is not the only cause for tissue dissemination.

### 3.2. The Consequences and Problems in Surgical Practice after Morcellation Banning

On the other hand, the utilization of the morcellation technique during myomectomy is not just a question of surgical practice, but it is a medical-legal issue, especially if an undiagnosed uterine LMS is morcellated and disseminated into the abdominal cavity during surgery. In addition, the loss of use of morcellation technique in gynecologic surgery led to the decrease in minimal invasive procedures and an increase of postoperative complications associated with open surgery.

Ottarsdottir et al. [28] reported a decrease of 40–60% in performing minimal-invasive procedures for UF in 2014 and 2015 compared with 2013. However, same authors observed that the majority of procedures each year were performed in a minimally invasive fashion and that the main factor associated with performing a minimally invasive hysterectomy was a well-trained surgeon.

Clark et al. [29] analyzed 77,637 hysterectomies and myomectomies and reported a 4% decrease in the use of laparoscopic hysterectomy for management of uterine leiomyomas after the FDA communication on morcellation and an increase in abdominal hysterectomy of 8% in the same period. Interestingly, there were no significant changes in the proportion of myomectomy compared with hysterectomy for the treatment of UFs.

It is known that laparoscopic myomectomy has advantages compared to open myomectomy. These advantages are well known, and it is not hard to assume the consequences of the decrease of laparoscopic practice after morcellation banning, especially laparoscopic myomectomy.

Tinelli et al. [30] showed the reduced intra and post-surgical blood loss and the application of pain relief medication for laparoscopic myomectomy, compared to open myomectomy. On the other hand, the duration of the operation is statistically longer in laparoscopic than in open myomectomy [30].

When comparing the laparoscopic and open myomectomy, it is important to emphasis a rate of possible recurrence of UF after surgery. Kotani et al. [31] compared the recurrence rate of a total of 474 patients who underwent laparoscopic myomectomy and 279 patients who underwent open myomectomy. They have observed a higher recurrence rate after laparoscopic myomectomy compared to open myomectomy, caused by more exhaustive extraction of smaller UF masses during the latter.

Ming et al. [32] performed a multicenter cohort study with a meta-analysis aimed to determine the risk of recurrence of UF after laparoscopic and open myomectomy. In a cohort study including 396 patients (83 patients who underwent laparoscopic and 313 patients who underwent open myomectomy), a similar recurrence rate was observed between laparoscopic and open myomectomy. Similar results were observed in the meta-analysis which include 2556 patients. However, they observed a statistically significant higher recurrence rate of UF after laparoscopic myomectomy than open myomectomy when patients had > 5 leiomyomas.

Chen et al. [33] conducted a meta-analysis that included twelve randomized clinical trials with a total of 1783 patients, to compare postoperative outcomes and complications in laparoscopic vs. open myomectomy. They observed a significant decrease in the blood loss, duration of postoperative ileus, and length of hospital stay after laparoscopic myomectomy. Conversely, they observed a longer duration of operation and higher medical cost after laparoscopic myomectomy compared to open myomectomy.

Multinu et al. [34] undertook a retrospective cohort study including 75,487 women to assess changes in the rates of 30-day major and minor complications of hysterectomy for benign gynecologic indications, after the FDA-issued statement. They observed the non-significant differences in major and minor complication rates before from 2013 through the first quarter of 2014) and after (from the fourth quarter of 2014 through 2015) the FDA-issued statement. However, in a subset of 25,571 women who underwent hysterectomy for UF, a significant increase in major and minor complications was observed after the FDA ban on morcellation. In this subgroup, a significant decrease in laparoscopic surgery was observed, with an increase in open abdominal surgery. Stentz et al. [35] examined the influence of the FDA communication on power morcellation on surgical approaches and morbidity after myomectomy. They collected data on about 3160 myomectomies between April 2012 and December 2013 (pre-FDA) and 4378 between April 2014 and December 2015 (post-FDA). The myomectomies performed post-FDA alert were more likely to be abdominal than laparoscopic, as compared with equal representations of laparoscopic and abdominal myomectomies in the pre-FDA era. The authors observed that abdominal myomectomy was associated with longer hospitalization, higher readmission, and greater morbidity in the pre-FDA, as well as post-FDA era. There was no difference in composite morbidity in different approaches to myomectomies in the pre-FDA and post-FDA eras.

Harris et al. [36] performed a retrospective cohort study to compare the incidence of major surgical, non-transfusion complications and postoperative period for hysterectomies performed before and after 15 months following the FDA statement on morcellation. The authors showed a significant decrease in laparoscopic hysterectomies and at the same time increase in open and vaginal myomectomies after the FDA ban. At the same time, there was an observed increase in major surgical complications and the rate of hospital readmission within 30 days.

### 3.3. The Morcellation Banning and Its Influence on Surgical Costs

It has been also observed that changes in laparoscopic practice after the FDA statement on morcellation led to an increase in costs, although eliminating morcellation from laparoscopic surgery is not cost-effective under a wide variety of probability and cost assumptions [37,38]. Rutstein et al. [39] developed a decision-analytic model comparing the cost-effectiveness of laparoscopic to open hysterectomy in terms of dollars. They observed that laparoscopic hysterectomy was less costly and yielded more QALYs (quality-adjusted life-year) in comparison with open hysterectomy for UF, when considering complications and morbidity as well as total direct hospital costs. Taking into account the rarity of occult LMS and the reduced incidence of operative and postoperative complications, laparoscopic hysterectomy with morcellation is a more cost-effective and less invasive alternative to open hysterectomy for the treatment of UF.

### 3.4. The Influence of Morcellation on Survey of Patients with Undiagnosed LMS

There are also conflicting findings concerning the influence of morcellation based on a survey of patients with preoperative undiagnosed LMS. On the one hand, it has been observed that morcellation did not affect disease-free survival of patients with LMS but, on the other, that patients with morcellation had 3-fold increased risk of death, in comparison with patients who did not have morcellation [40].

Lin et al. [41] reported there is no significant difference in recurrence rate, as well as disease-free survival and overall survival in patients, diagnosed with LMS stage 1, with morcellation compared with patients without morcellation. This finding implies that survival of leiomyosarcoma stage 1 patients depends on tumor size, not on morcellation utilization.

Gao et al. [42] also observed that there is no difference in recurrence-free survival and overall survival in patients treated for UF, with a postoperative diagnosis of uterine LMS. However, the 5-year overall survival and recurrence-free survival were lower in patients with morcellation compared to patients without morcellation. The grade level of LMS, but not the morcellation, significantly influenced the prognosis of patients diagnosed for uterine LMS.

Taking into account the advantages of laparoscopic approaches in myomectomy and, however, the factors which influence the survey of patients with preoperative undiagnosed LMS, the international gynecologic societies as well as the European Society of Gynecological Oncology (ESGO), the International Society for Gynecologic Endoscopy (ISGE), the European Society for Gynecological Endoscopy (ESGE), Die Deutsche Gesellschaft für Gynäkologie und Geburtshilfe (DGGG) and the American College of Obstetricians and Gynecologists (ACOG) have recommended statements about morcellation [43,44,45,46,47,48].

The common recommendations are that all patients must be assessed for individual risk of LMS and benefit from the possibility of a laparoscopic approach and morcellation utilization [49]. In patients who are suspected that having LMS or diagnosed for LMS, the morcellation is to be avoided [50]. Most of the societies do not ban morcellation and laparoscopic hysterectomy/myomectomy per se, but suggest that decisions should reflect the clinical characteristics of patients. It is also recommended that each patient must be informed about the risks and benefits of open or laparoscopic hysterectomy/myomectomy and morcellation and give a clear preoperative informed consent.

## 4. Conclusions

Nowadays, in the light of what has been highlighted in the literature, there are no particular needs or requirements to modify one’s surgical behavior concerning the morcellation of UFs, as long as it is consistent with the assessment of the patients’ characteristics and risk/benefit ratio. We believe that power morcellation can still be widely used, after selecting the patients to submit to laparoscopy. Of course, it is necessary to use the usual prudence before undertaking a myomectomy, not least to avoid annoying and expensive medical-legal ligation in the courts.

## Figures and Tables

**Figure 1 healthcare-10-02087-f001:**
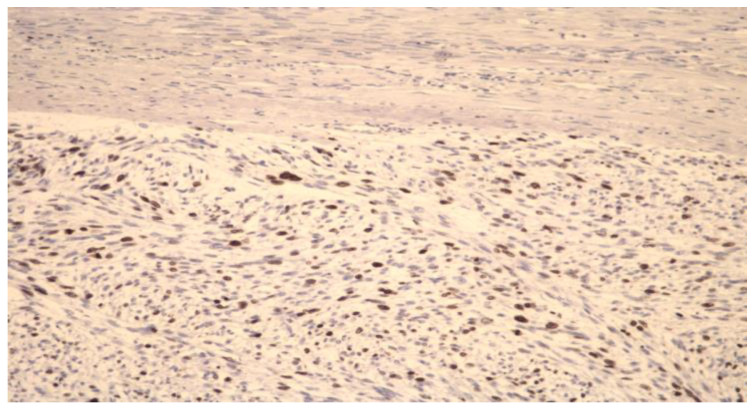
Immunohistochemical preparation of LMS cells (magnification at 10×). It could be seen that the tumor is heterogeneous. The tumor cells frequently express Ki-67 antigen.

**Figure 2 healthcare-10-02087-f002:**
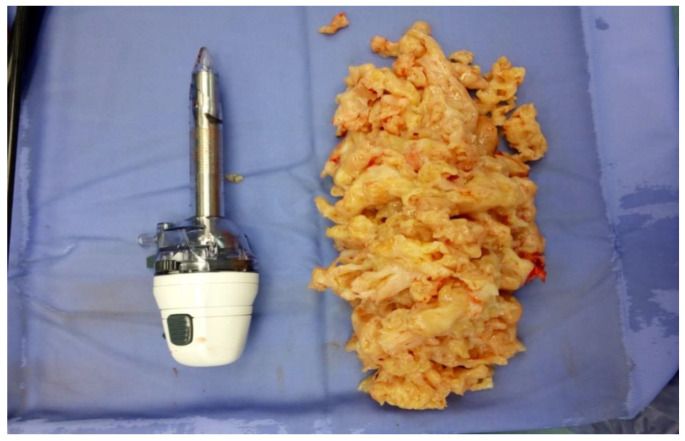
The morcellated uterine fibroid.

**Table 1 healthcare-10-02087-t001:** Incidence and risk for occult uterine leiomyosarcoma (LMS).

Study of Incidence of Occult Uterine LMS	Incidence or Risk for Occult Uterine LMS
Bannettet et al. [12]	125/34,728
Tan et al. [13]	0.27%
Chen et al. [14]	0.54%
Kundu et al. [15]	2/2825
Valzacchi et al. [16]	2.15/1000
Tchartchian et al. [17]	0/1498
Gitas et al. [18]	4/1683
Pados et al. [19]	0%

## Data Availability

Not applicable.

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
