# Peer review of "Fibroid Removal after Myomectomy: An Overview on the Problems of Power Morcellation"

_healthcare, 2022, doi:10.3390/healthcare10102087_

Round 1
Reviewer 1 Report
Dear Authors,
this manuscript concerns an interesting topic which is the use of power morcellation for fibroid removal after minimally invasive myomectomy. The main issue with this article is its methodology.
The authors in this manuscript claim that this review is narrative, however, within the article there are components of both a narrative and a systematic review that do not fit. Firstly, the aim of this review is not clear enough. Furthermore, although this review is narrative, the materials and methods are described as this article concerns a systematic review. In a narrative review there is no reason for having a flowchart. Furthermore, the authors claim in the flowchart that the studies included in the review are 46. Nonetheless, I could not identify any 46 studies in the results section that were included in this paper.
Moreover, the authors have divided the Results section into two paragraphs (3.1. The incidence of uterine neoplasms in women who underwent laparoscopic surgery with 100 morcellation - 3.2. The incidence of occult un-expected uterine malignancies in specimens retrieved by morcellation during laparoscopic surgery.) Personally, I could not find any differences between the two paragraphs. For instance, the 3.2 section is named "The incidence of occult un-expected uterine malignancies.." However, this applies to the 3.1 section too. Therefore, I could not understand which studies were included in the one or the other section and why.
The Discussion section although really detailed, is really long. The authors instead of commenting on the results, they have just mentioned more studies, which could have been included in the Results section.
Finally, the conclusion section should be modified, since it is very strong. Although what the authors claim might be right, these are not well supported by the findings of the study. For instance, I could not figure out how the authors reached the conclusion that "in the light of what has been highlighted in the literature, there are no particular needs or requirements to modify one's surgical behavior regarding the morcellation of UFs."
Unfortunately, I believe that this manuscript could not be published in its current form. However, instead of rejecting it, I would be happy to re-review this manuscript after a really MAJOR revision. If the authors decide that they want to continue with a narrative review, I suggest that they change the format of the paper. There is no reason for having such a methodology, nor describing the results and the discussion in this manner. Instead, they should have both the results and the discussion in the same section and re-organize the whole paper. This could be done in a way such that the paper starts with an argument and the authors try to elaborate and explain it throughout the main body of the manuscript, and finally reach a conclusion around it based on what has been said before.
Author Response
Dear Authors,
this manuscript concerns an interesting topic which is the use of power morcellation for fibroid removal after minimally invasive myomectomy. The main issue with this article is its methodology. The authors in this manuscript claim that this review is narrative, however, within the article there are components of both a narrative and a systematic review that do not fit.
Answer: We reorganized the materials and methods section, to better fit the manuscript with the characteristic of a narrative review.
Firstly, the aim of this review is not clear enough. Furthermore, although this review is narrative, the materials and methods are described as this article concerns a systematic review.
Answer: according to reviewer comment, we edited the materials and methods paragraph, changing the setup of the M&M.
In a narrative review there is no reason for having a flowchart. Furthermore, the authors claim in the flowchart that the studies included in the review are 46.
Answer: we modified the layout of the manuscript according to the editorial rules, in accordance with what the reviewer pointed out; our paper is narrative review, therefore we deleted flowchart, even if we included 48 papers in the narrative review.
Nonetheless, I could not identify any 46 studies in the results section that were included in this paper.
Answer: currently 48 research papers have been included in the narrative review. Your comment helped us to reorganize paper aimed it will be more comprehensible. However, beside the paper which reports the result about incidence and risk for occult leiomyosarcoma in patients diagnosed for uterine fibroids, in this narrative review are included also papers which are used for the explanation of our standpoint about per se morcellation banning.
Moreover, the authors have divided the Results section into two paragraphs (3.1. The incidence of uterine neoplasms in women who underwent laparoscopic surgery with 100 morcellation - 3.2. The incidence of occult un-expected uterine malignancies in specimens retrieved by morcellation during laparoscopic surgery.) Personally, I could not find any differences between the two paragraphs.
Answer: accordingly with reviewer comments’, we have merged two paragraph in one paragraph which is divided into some subheadings, explaining step by step problems of banning of morcellation of uterine fibroid.
For instance, the 3.2 section is named "The incidence of occult un-expected uterine malignancies.." However, this applies to the 3.1 section too. Therefore, I could not understand which studies were included in the one or the other section and why.
Answer: Accordingly, we merged two paragraph in one paragraph and explained, in the subheadings, the main consequences of morcellation banning and influence of surgical practice and outcomes.
The Discussion section although really detailed, is really long. The authors instead of commenting on the results, they have just mentioned more studies, which could have been included in the Results section.
Answer: we reorganized the results and discussion section, to make it more readable. We hope these changes make our manuscript more comprehensive and clearer.
Finally, the conclusion section should be modified, since it is very strong. Although what the authors claim might be right, these are not well supported by the findings of the study.
Answer: we edited the results and discussion section basing on your suggestions, explaining throughout the section our standpoint.
For instance, I could not figure out how the authors reached the conclusion that "in the light of what has been highlighted in the literature, there are no particular needs or requirements to modify one's surgical behavior regarding the morcellation of UFs."
Answer: Accordingly, we have modified these sentences in “Nowadays, in the light of what has been highlighted in the literature, there are no particular needs or requirements to modify one's surgical behavior concerning the morcellation of uterine fibroids as long as it is consistent with the assessment of the patients’ characteristics and risk/benefit ratio.”
Unfortunately, I believe that this manuscript could not be published in its current form. However, instead of rejecting it, I would be happy to re-review this manuscript after a really MAJOR revision. If the authors decide that they want to continue with a narrative review, I suggest that they change the format of the paper. There is no reason for having such a methodology, nor describing the results and the discussion in this manner. Instead, they should have both the results and the discussion in the same section and re-organize the whole paper. This could be done in a way such that the paper starts with an argument and the authors try to elaborate and explain it throughout the main body of the manuscript, and finally reach a conclusion around it based on what has been said before.
Answer: As well as we wrote, the results and discussion are now merged in the same section and explain our arguments throughout the section. To make the manuscript more readable and clearer, we introduce subheadings into results and discussion subsection, with our comments in all subheadings. We hope that these changes improved our manuscript and make it suitable for publication.
Reviewer 2 Report
Thank you for the opportunity to review the manuscript entitled “Fibroid removal after myomectomy: an overview on the problem of power morcellation“ .
Since uterine sarcomas are most commonly diagnosed following surgery the authors reviewed the risk of tissue dissemination in patients operated on for presumed benign leiomyomas who are found postoperatively to have uterine sarcoma.
To become interesting enough to attract the readers’ attention: the review needs a specific discussion on the diagnostic process for differentiating preoperatively a leiomyoma from a uterine sarcoma.
Some Specific Comments:
1. Please explain all abbreviations used in manuscript.
2. What is relevance of the of Figure 2 and Figure 3? Please explain.
Author Response
Thank you for the opportunity to review the manuscript entitled “Fibroid removal after myomectomy: an overview on the problem of power morcellation“.
Since uterine sarcomas are most commonly diagnosed following surgery the authors reviewed the risk of tissue dissemination in patients operated on for presumed benign leiomyomas who are found postoperatively to have uterine sarcoma.
To become interesting enough to attract the readers’ attention: the review needs a specific discussion on the diagnostic process for differentiating preoperatively a leiomyoma from a uterine sarcoma.
Answer: We included paragraph about problem of diagnostic process for differentiating preoperatively a leiomyoma from a uterine sarcoma. In the same time, this paragraph explains the diagnostic background of potentially undiagnosed uterine leiomyosarcoma.
Some Specific Comments:
- Please explain all abbreviations used in manuscript.
- What is relevance of the of Figure 2 and Figure 3? Please explain.
Answer: We checked the manuscript and explained all used abbreviation. The Figure 2 and Figure 3 are obtained from practice of authors and pathological basis of authors clinic, aimed to the give insights about morcellation and pathological diagnostics the all readers of paper, not only those from field of gynecology.
Round 2
Reviewer 1 Report
Dear Authors,
I was very satisfied to read the revision of your manuscript. I was really glad to see that the comments and suggestions I made were addressed. Based on the new format of the manuscript and the change in the materials and methods section, as well as the new results and discussion section, I recommend that this manuscript is published unaltered.
Reviewer 2 Report
I am satisfied to see that the comments and suggestions I made were addressed. Based on the new format of the manuscript I recommend that this manuscript is published unaltered.